# Prevalence characteristics of cervical human papillomavirus infection in Chengdu and Aba District, Sichuan Province, China

Qianqian Wang[1]*, Min Xu[1], Hua Zhou[2], Yahui Li[1], Jichun Ma[2], Xuan Zhu[1], Weijun He[1]

1 Department of Clinical Laboratory, Chengdu Women's and Children's Central Hospital, School of Medicine, University of Electronic Science and Technology of China, Chengdu, China, 2 Department of Clinical Laboratory, Aba Maternal and Child Health Hospital, Aba, China

☯ These authors contributed equally to this work.
* wangqianqian0330@126.com

## Abstract

### Purpose

The genotype distribution of human papillomavirus (HPV) infection varies greatly in different regions. This study aims to determine the prevalence and type-specific distribution of HPV among females from Chengdu and Aba in Sichuan Province, which differ in geographical location, economic status, and living habits. These can serve as evidence of epidemic patterns for future design and implementation of vaccination and screening programs.

### Methods

A retrospective cross-sectional study was conducted on 144 113 women who underwent cervical screening at Chengdu Women's and Children's Central Hospital from January 2015 to September 2020. Meanwhile, 1799 samples from February 2018 to December 2021 were collected from Aba Maternal and Child Health Hospital. HPV DNA genotype testing was performed using real-time PCR. The overall prevalence, annual trend, age-specific prevalence, and type distribution were analyzed.

### Results

The overall HPV prevalence was 22.51% in Chengdu. During 2015–2020, the highest prevalence rate was observed in 2018. Age-specific HPV distribution displayed a bimodal distribution among women aged ≤25 or ≥46 years old. The top three prevalent genotypes were HPV52, -16, and -58. Although the total prevalence of HPV in Aba was 14.23%, there was an upward trend from 2018 to 2021. However, no significant differences were identified in HPV infection rate across all age groups. HPV52, -53, and -16 were the major genotypes. Furthermore, single-type HPV infections and high-risk HPV infections were identified as the most common infection types in both regions.

**Data Availability Statement:** All relevant data are within the paper and its Supporting Information files.

**Funding:** The author(s) received no specific funding for this work.

**Competing interests:** The authors have declared that no competing interests exist.

## Conclusion

Our findings demonstrate the overall prevalence of HPV was still high in Chengdu and Aba. The age-specific prevalence distribution demonstrated different patterns. Non-vaccine-covered HR-HPV53, -51and LR-HPV81, -CP8304 were frequently detected, which was worth significant clinical attention. In summary, regional HPV screening provides valuable clinical guidance for cervical cancer prevention and vaccine selection in Western China.

## Introduction

Infection of human papillomavirus (HPV) is a great threat to women's health worldwide. There are over 200 HPV types recognized based on DNA sequence data, classified into high-risk and low-risk HPV genotypes by their carcinogenicity. Most low-risk genotypes are benign. Only a small proportion of infections with certain types persist and progress to cancer, such as oropharyngeal, cervical, vulvar, vaginal, and penile cancer. Cervical cancer is by far the most common HPV-related disease [1].

In 2020, an estimated 604,000 new cases and 342,000 deaths were attributed to cervical cancer which was the fourth most frequently diagnosed cancer and the fourth leading cause of cancer death in women [2]. Both incidence and mortality exhibited an increase compared to 2018 [3]. Furthermore, developing countries saw a higher death rate than developed ones. [2]. It represented approximately 18% of the global cervical cancer deaths in China, with the sixth-highest incidence and seventh-highest death cases [4].

Nearly 99% of cervical cancer cases are associated with HPV infection worldwide [5]. The global prevalence of high-risk HPV infection is 10.4% [6], and in some developing countries, it can be as high as 36.5% [7, 8]. In contrast to most malignant tumors, cervical cancer is thought to be nearly completely preventable, benefiting from the highly effective primary (vaccination) and secondary (screening) preventative strategies. [2]. HPV vaccination programs can protect from HPV infection and then reduce the future burden of invasive cervical cancer [9]. High-quality screening strategies, including visual inspection with acetic acid, cervical cytology, and HPV testing, are critical for unvaccinated women and those infected with non-vaccine HPV types. However, the implementation of these two means varies inequitably across countries and regions [2]. In China, four HPV vaccines have been approved since 2016. However, the promotion of vaccination remains limited and challenged due to high prices and insufficient supply of vaccines [10].

Considering the shortage of HPV vaccine prevention coverage and distribution difference of type-specific HPV infection, studies have been conducted to investigate the prevalence and incidence of HPV genotypes [11], as well as to evaluate whether the current vaccines and screening strategies could meet the public health needs in different regions of China [12, 13]. To date, the majority of these endeavors have been focused on developed regions in eastern and southern China [14–16], data regarding western China remains less. In particular, there is a lack of data from remote high-altitude locations. Therefore, this study was designed to present a retrospective analysis of the epidemiological characteristics of HPV infections in Chengdu and Aba, both in Sichuan Province, China, representing well-developed big cities and underdeveloped regions in Tibet Plateau respectively. We sought to reveal the overall prevalence, age-specific prevalence, genotype distribution, and trend of HPV infections with massive long-term data from these two regions. These findings attempted to provide a general

insight into the HPV burden in western China and serve as evidence of epidemic patterns for future design and implementation of vaccination and screening programs.

## Materials and methods

### Ethics statement

The Ethics Committee of Chengdu Women's and Children's Central Hospital approved this project (Lot No.: 2023(66)). The management and publication of patient information in this research were strictly following the Declaration of Helsinki, including confidentiality and anonymity. The ethics committee waived the requirement for informed consent. We started to conduct our study on July 1, 2023, after we obtained the ethical approval from the ethics committee on June 25, 2023.

### Sample collection

144 113 samples were collected from women who underwent gynecological examination and participated in HPV screening at Chengdu Women's and Children's Central Hospital from January 2015 to September 2020. Patients from the outpatient department and physical examination center were included in our study. The HPV genotyping results and relevant clinical information were all recorded, including age and clinical diagnosis available in the database. Cervical cell samples were obtained by using a specialized cervical swab, which were then immersed in a saline medium for HPV genotype testing. Meanwhile, 1799 samples from February 2018 to December 2021 were collected from female participants in the cervical cancer screening program at Aba Maternal and Child Health Hospital and subsequently delivered to Gaoxin Daan Medical Laboratory for further testing. Relevant information regarding age and region was also extracted.

### DNA extraction and genotyping

HPV genotyping diagnosis kit (Yaneng Bio (Shenzhen) Co., Ltd.) was used for DNA extraction and genotyping at Chengdu Women's and Children's Central Hospital. Firstly, DNA was extracted and amplification by polymerase chain reaction (PCR) was performed. The PCR parameters were as follows: 50°C for 15min, 95°C for 10min, 40 amplification cycles (denaturation at 94°C for 30s, annealing at 42°C for 90s, extension at 72°C for 30s) and 72°C for 5min). Subsequently, genotyping was performed by hybridization and gene microarray technology on gene chips fixed with HPV-type-specific probes. Quality controls were simultaneously involved in the experiment by using positive and negative controls provided within the kit. A total of 23 HPV genotypes were identified, including 17 high-risk types (16, 18, 31, 33, 35, 39, 45, 51, 52, 53, 56, 58, 59, 66, 68, 73 and 82) and 6 low-risk types (6, 11, 42, 43, 81 and 83). The genotyping of samples from Aba was conducted by Gaoxin Daan Medical Laboratory using reverse dot blot-PCR assay, which classified 19 HPV genotypes, consisting of 15 HR-HPV types (16, 18, 31, 33, 35, 39, 45, 51, 52, 53, 56, 58, 59, 66 and 68) and 4 LR-HPV types (CP8304, 6, 11 and 43).

### Statistical analysis

All statistical analyses were operated by using SPSS 22.0(SPSS Inc., Chicago, IL, USA). HPV infection prevalence and type-specific distribution in both total and designated groups were calculated. Rates and trends of HPV infection among groups divided according to age, year, or region were analyzed. The chi-squared test was employed to evaluate the significance of

differences between groups. A two-sided P-value less than 0.05 (P < 0.05) was considered statistically significant.

## Results

### Overall prevalence, genotype distribution, and annual prevalence of HPV in different regions

A total of 144 113 samples were collected and analyzed by HPV genotyping in Chengdu from 2015 to 2020. The genotype testing showed 32 443 samples were HPV positive, resulting in an overall prevalence rate of 22.51% (Table 1), HPV52 (5.04%) was the most prevalent followed by HPV16 (3.12%), -58 (2.63%), -81 (2.46%), -53 (2.28%) and -51 (2.02%) (Fig 1). Meanwhile, among HR-HPV types, HPV52, -16, and -58 remained the top three prevalent genotypes, while its counterpart LR-HPV was HPV81, -42 and -43 respectively (Table 1). We observed significant differences in the prevalence of single, multiple, and total infections from 2015 to 2020. All three categories exhibited a peak in 2018, while the curve for multiple infections remained relatively flat. Of the six top genotypes, HPV16 displayed a similar trend as that seen in the total infection rate (Fig 2).

Meanwhile, among 1799 women in Aba from 2018 to 2021, the prevalence of HPV infection was found to be 14.23%, with 256 cases positive (Table 2). Notably, HPV52 (3.45%), -53

**Table 1. Overall prevalence of HPV genotype in single and multiple infections in Chengdu.**

| | Genotypes | Single infection No. (%) | Multiple infections No. (%) | Total No. (%) |
|---|---|---|---|---|
| **HR-HPV**[*] | 52 | 4399 (3.05) | 2859 (1.98) | 7258 (5.04) |
| | 16 | 2540 (1.76) | 1951 (1.35) | 4491 (3.12) |
| | 58 | 2167 (1.50) | 1617 (1.12) | 3784 (2.63) |
| | 53 | 1766 (1.23) | 1524 (1.06) | 3290 (2.28) |
| | 51 | 1542 (1.07) | 1371 (0.95) | 2913 (2.02) |
| | 68 | 1045 (0.73) | 893 (0.62) | 1938 (1.34) |
| | 56 | 915 (0.63) | 907 (0.63) | 1822 (1.26) |
| | 59 | 865 (0.60) | 957 (0.66) | 1822 (1.26) |
| | 18 | 904 (0.63) | 800 (0.56) | 1704 (1.18) |
| | 33 | 664 (0.46) | 712 (0.49) | 1376 (0.95) |
| | 66 | 615 (0.43) | 717 (0.50) | 1332 (0.92) |
| | 39 | 634 (0.44) | 642 (0.45) | 1276 (0.89) |
| | 31 | 430 (0.30) | 423 (0.29) | 853 (0.59) |
| | 35 | 292 (0.20) | 384 (0.27) | 676 (0.47) |
| | 45 | 223 (0.15) | 229 (0.16) | 452 (0.31) |
| | 73 | 95 (0.07) | 135 (0.09) | 230 (0.16) |
| | 82 | 107 (0.07) | 107 (0.07) | 214 (0.15) |
| **LR-HPV**[#] | 81 | 1863 (1.29) | 1686 (1.17) | 3549 (2.46) |
| | 42 | 788 (0.55) | 942 (0.65) | 1730(1.20) |
| | 43 | 813 (0.56) | 844 (0.59) | 1657 (1.15) |
| | 6 | 591 (0.41) | 738 (0.51) | 1329 (0.92) |
| | 11 | 439 (0.30) | 383 (0.27) | 822 (0.57) |
| | 83 | 79 (0.05) | 115 (0.08) | 194 (0.13) |
| | Total | 23783 (16.50) | 8660 (6.01) | 32443 (22.51) |

Single infection versus multiple infections considering HR-HPV[*] P< 0.001 and LR- HPV[#] P = 0.154. HPV, Human papillomavirus; HR-HPV, High-risk HPV; LR-HPV, Low-risk HPV.

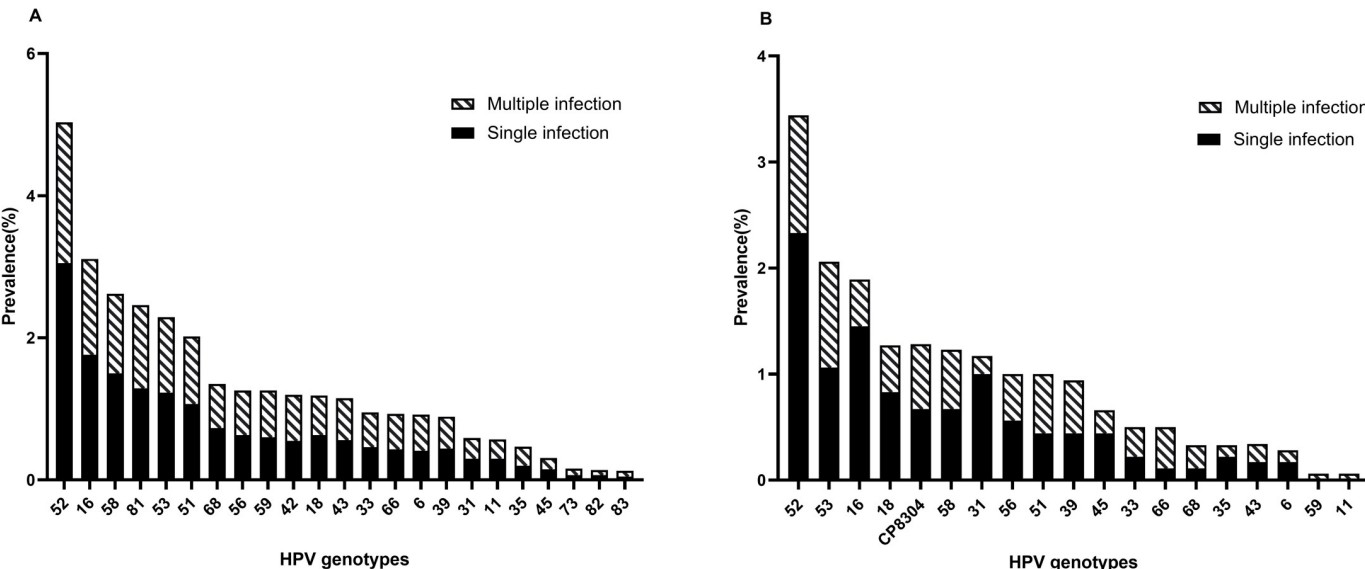

**Fig 1. Genotype-specific distribution of HPV infections.** (A) The six top prevalent HPV genotypes in Chengdu were HPV52 (5.04%), −16 (3.12%), −58 (2.63%), −81 (2.46%), −53 (2.28%), and −51(2.02%); (B) The six top prevalent HPV genotypes in Aba were HPV52 (3.45%), −53 (2.06%), −16 (1.83%), −18 (1.28%), −CP8304 (1.28%), and −58(1.22%).

(2.06%), -16(1.83%), -18 (1.28%), -CP8304 (1.28%) and -58 (1.22%) were the top six prevalent genotypes (Fig 1). Among HR-HPV genotypes, HPV 52, -53, and -16 were the top three prevalent, while HPV CP8304, -43, and -6 were the top three among LR-HPV (Table 2). Besides, the overall trend indicated an increase in single, and total prevalence rates from 2018 to 2021. However, there were no significant differences among these groups regarding the top six prevalent genotypes (Fig 3).

## Prevalence of LR-HPV, HR-HPV, and single, multiple HPV infections in different regions

Overall, the positive rates for LR-HPV, HR-HPV, and mixed LR- and HR-HPV were 3.33%, 16.56%, and 2.62% respectively in Chengdu (P<0.001). The prevalence of single infection was

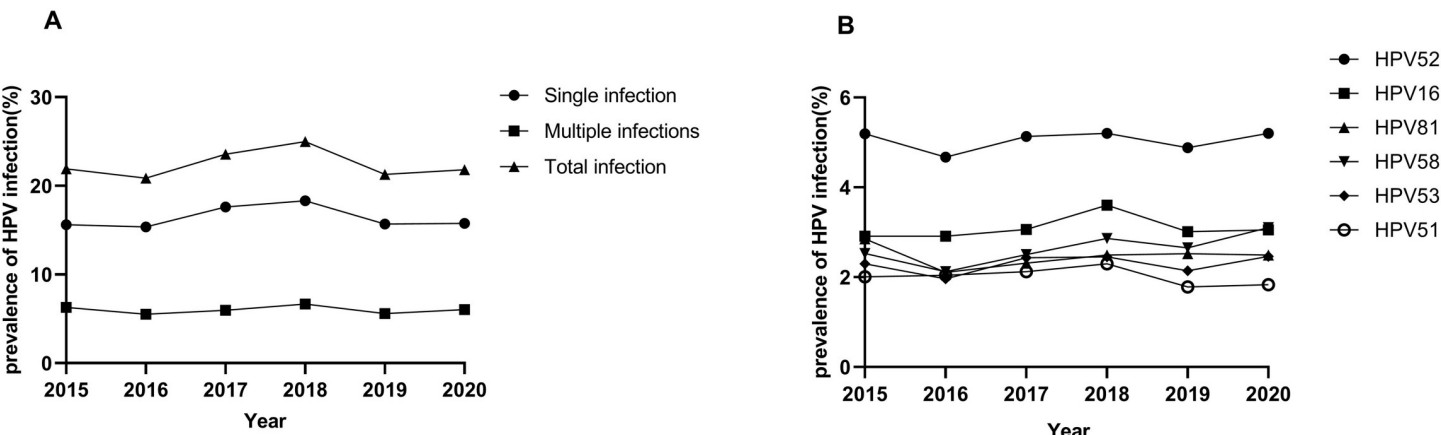

**Fig 2. Prevalence of HPV infections in Chengdu from 2015 to 2020(%).** (A) Prevalence of HPV genotypes in single, multiple and total infections; (B) Changes in the prevalence of the top six HPV genotypes.

**Table 2. Overall prevalence of HPV genotype in single and multiple infections in Aba.**

| | Genotypes | Single infection No. (%) | Multiple infections No. (%) | Total No. (%) |
|---|---|---|---|---|
| HR-HPV[*] | 52 | 42 (2.33) | 20 (1.11) | 62 (3.45) |
| | 53 | 19 (1.06) | 18 (1.00) | 37 (2.06) |
| | 16 | 26 (1.45) | 8 (0.44) | 33 (1.83) |
| | 18 | 15 (0.83) | 8 (0.44) | 23 (1.28) |
| | 58 | 12 (0.67) | 10 (0.56) | 22 (1.22) |
| | 31 | 18 (1.00) | 3 (0.17) | 21 (1.17) |
| | 56 | 10 (0.56) | 8 (0.44) | 18 (1.00) |
| | 51 | 8 (0.44) | 10 (0.56) | 18 (1.00) |
| | 39 | 8 (0.44) | 9 (0.50) | 17 (0.94) |
| | 45 | 8 (0.44) | 4 (0.22) | 12 (0.67) |
| | 33 | 4 (0.22) | 5 (0.28) | 9 (0.50) |
| | 66 | 2(0.11) | 7 (0.39) | 9 (0.50) |
| | 68 | 2 (0.11) | 4 (0.22) | 6 (0.33) |
| | 35 | 4 (0.22) | 2 (0.11) | 6 (0.33) |
| | 59 | 0 | 1 (0.06) | 1 (0.06) |
| LR-HPV[#] | CP8304 | 12 (0.67) | 11 (0.61) | 23 (1.28) |
| | 43 | 3 (0.17) | 3 (0.17) | 6 (0.33) |
| | 6 | 3 (0.17) | 2 (0.11) | 5 (0.28) |
| | 11 | 0 | 1 (0.06) | 1 (0.06) |
| | Total | 196 (10.89) | 60 (3.34) | 256 (14.23) |

Single infection versus multiple infections considering HR-HPV[*] $P< 0.05$ and LR- HPV[#] $P = 0.865$.

16.50%, accounting for 73.31% of the total instances of HPV positivity (32443). In contrast, the prevalence of multiple infections was only 6.01%, meanwhile, double infection played a dominant role in cases with multiple infections (Table 3).

In Aba, the LR-, HR- and mixed LR- and HR-HPV positive rates were 1.00%, 12.34%, and 0.89% respectively. The prevalence of single infection was 10.89%, accounting for 76.56% of

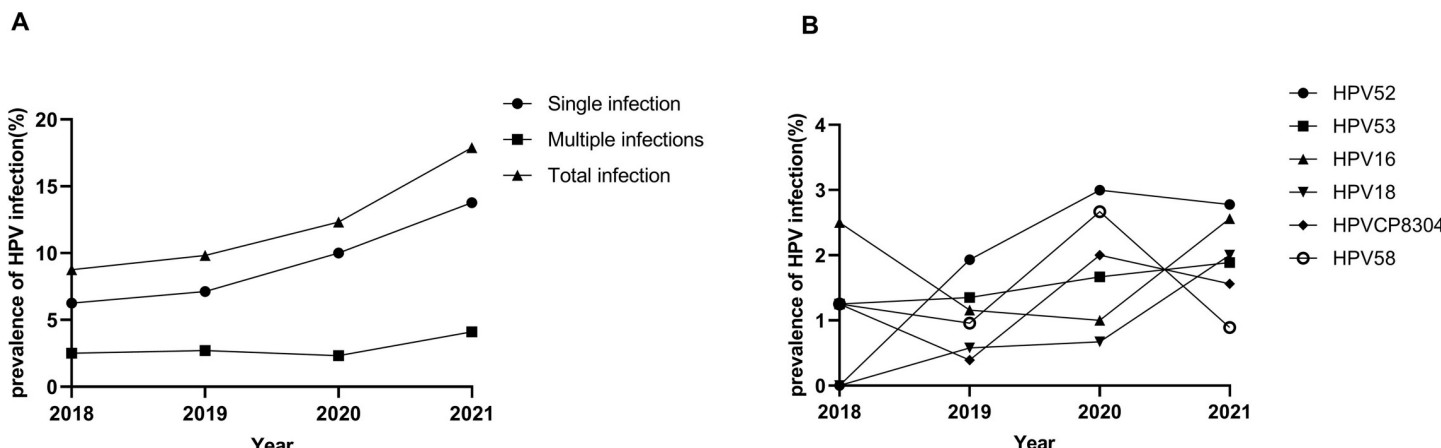

**Fig 3. Prevalence of HPV infections in Aba from 2018 to 2021(%).** (A) Prevalence of HPV genotypes in single, multiple and total infections; (B) Changes in the prevalence of the top six HPV genotypes.

**Table 3. Age-specific distribution of HPV infection in Chengdu.**

| | Age (years) No. (%) | | | | | P |
|---|---|---|---|---|---|---|
| | ≤25 (n = 18566) | 26–35 (n = 77631) | 36–45 (n = 28718) | ≥46 (n = 19198) | Total (n = 144113) | |
| **HPV positive numbers** | 5167 (27.83) | 16088 (20.72) | 6245 (21.75) | 4943 (25.75) | 32443 (22.51) | <0.001 |
| **Number of co-infections** | | | | | | |
| 1 HPV genotype | 3291 (17.73) | 12081 (15.56) | 4924 (17.15) | 3487 (18.16) | 23783 (16.50) | <0.001 |
| 2 HPV genotypes | 1196 (6.44) | 2978 (3.84) | 1006 (3.50) | 1021 (5.32) | 6201 (4.30) | <0.001 |
| 3 HPV genotypes | 440 (2.37) | 746 (0.96) | 231 (0.80) | 288 (1.50) | 1705 (1.18) | <0.001 |
| 4 HPV genotypes | 146 (0.79) | 195 (0.25) | 70 (0.24) | 88 (0.46) | 499 (0.35) | <0.001 |
| ≥ 5 HPV genotypes | 94 (0.51) | 88 (0.11) | 14 (0.05) | 59 (0.31) | 255 (0.18) | <0.001 |
| **HPV genotype** | | | | | | |
| HR-HPV only | 3564 (19.20) | 11984 (15.44) | 4719 (16.43) | 3602 (18.76) | 23869 (16.56) | <0.001 |
| LR-HPV only | 740 (3.99) | 2400 (3.09) | 977 (3.40) | 679 (3.54) | 4796 (3.33) | <0.001 |
| mixed LR- and HR-HPV | 863 (4.65) | 1704 (2.19) | 549 (1.91) | 662 (3.45) | 3778 (2.62) | <0.001 |

the 256 positive cases. Multiple infections (3.34%) were rare, with double infection being the most common (Table 4).

## Age-specific Prevalence of HPV infection in different regions

According to the age group, participants were categorized into four groups: ≤ 25, 26–35, 36–45, and ≥46 years. The HPV prevalence curve in Chengdu demonstrated a bimodal distribution, with the first peak observed in the ≤25 years group (27.83%), followed by a sharp decline in the 26–35 group (20.72%), then increased to reach a second peak of 25.75% among those aged ≥46. Notably, the rates of LR- and HR-HPV-only infection exhibited a similar trend. The mixed infection rate showed a bimodal pattern with the same peaks, while the lowest rate was found in the 36–45 years group. The trend of single infection rate followed the same pattern as that of overall HPV infection, with the highest incidence observed in individuals aged ≥46 years. Interestingly, the trend of dual and multiple infection rates exhibited a similarity to that

**Table 4. Age-specific distribution of HPV infection in Aba.**

| | Age(years) No. (%) | | | | | P |
|---|---|---|---|---|---|---|
| | ≤25 (n = 58) | 26–35 (n = 347) | 36–45 (n = 615) | ≥46 (n = 779) | Total (n = 1799) | |
| **HPV positive numbers** | 9 (15.52) | 55 (15.85) | 85 (13.82) | 107 (13.74) | 256 (14.23) | = 0.78 |
| **Number of co-infections** | | | | | | |
| 1 HPV genotype | 5 (8.62) | 43 (12.39) | 68 (11.06) | 80 (10.27) | 196 (10.89) | = 0.70 |
| 2 HPV genotypes | 3 (5.17) | 11 (3.17) | 14 (2.28) | 23 (2.95) | 51 (2.83) | = 0.57 |
| 3 HPV genotypes | NA | 1 (0.29) | 2 (0.33) | 3 (0.39) | 6 (0.33) | = 0.96 |
| 4 HPV genotypes | NA | NA | 1 (0.16) | 1 (0.13) | 2 (0.11) | = 0.89 |
| ≥5 HPV genotypes | 1 (1.72) | NA | NA | NA | 1 (0.06) | = 0.08 |
| **HPV genotype** | | | | | | |
| HR-HPV only | 6 (10.34) | 48 (13.83) | 74 (12.03) | 93 (11.94) | 221 (12.28) | = 0.78 |
| LR-HPV only | 1 (1.72) | 5 (1.44) | 5 (0.81) | 7 (0.9) | 18 (1.00) | = 0.73 |
| mixed LR- and HR-HPV | 2 (3.45) | 2 (0.58) | 6 (0.98) | 7 (0.9) | 17 (0.94) | = 0.22 |

of mixed infection (Table 3). However, no significant differences were observed in the HPV infection rates across the four age groups in Aba (Table 4).

## Age distribution of dominant types

Based on the aforementioned findings, the six most prevalent HPV genotypes in Chengdu were as follows: HR-HPV genotype -52, -16, -58, -53, -51, and LR-HPV -81. Further investigations into the distribution features of these six genotypes across age groups were conducted. For -52 and -16, the highest infection rates were found in cases aged ≤25 years. Subsequently, the incidences decreased to the lowest point in those aged 26–35 and 36–45 years, followed by an obvious increase in individuals ≥46 years. Interestingly, despite the similar patterns observed in the trends of -58, -53, and -81, the most prevalent age groups were ≥46 years. Genotype -51 exhibited a peak incidence among cases aged ≤25, with no significant differences observed among other age groups (Fig 4). According to these results, women aged ≤25 years constitute the vast majority of individuals affected by these six dominant HPV infections.

In Aba, the six most prevalent genotypes were HR-HPV -52, -53, -16, -18, -58, and LR-HPV CP8304. However, there was no statistically significant difference in the prevalence of these genotypes across four age groups (Fig 4).

## Discussion

The considerable variation in HPV prevalence based on geography and population has prompted many investigations into regional epidemic strategies. Disparities also exist within countries or regions. Our study consisted of two groups located in different geographical areas, with varying economic levels and living habits, despite both being situated in Sichuan Province. The overall HPV prevalence in the two distinct regional groups showed 22.51% in Chengdu and 14.23% in Aba.: A meta-analysis of HPV prevalence in cytologically normal women from five continents showed that the global HPV prevalence was 11.7%. The continents, in descending order of prevalence, were: Africa (21.1%), Europe (14.2%), America (11.5%), and Asia (9.4%). Eastern Asia (10.7%) had the second highest infection rate in Asia following Southeastern Asia (14.0%) [17].Previous reports have indicated that rates of HPV positivity range from 6.70 to 44.50% in China [18]. The overall HPV infection of Chengdu was similar to that reported in Hangzhou (22.3%) [19], Taizhou area (22.8%) [20] and Guangdong (21.06%) [21], lower than in Jiangsu (26.92%) [22], Jilin (34.40%) [23], Shandong (28.4%) [24],

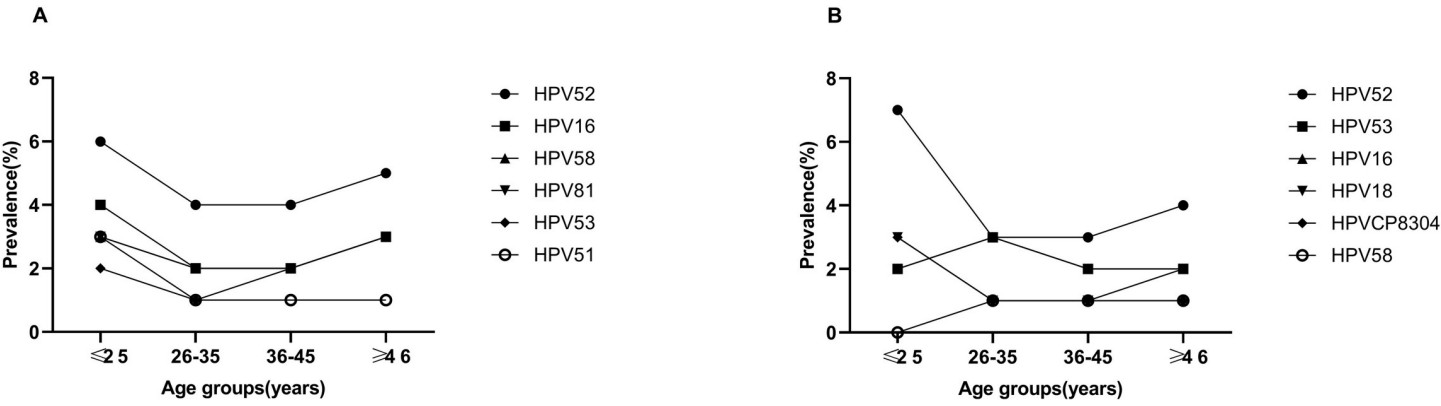

**Fig 4. The prevalence of six most prevalent HPV types by age groups.** (A) The epidemic characteristics in Chengdu. (B) The epidemic characteristics in Aba.

but higher than in Chongqing (18.59%) [25], Shanghai (17.92%) [26], Beijing (8.22%) [27]. The overall HPV infection of Aba was consistent with that reported in Xinjiang (14.02%) [28], southern Hunan (14.59%) [29], lower than in Tibet Autonomous Region (28.14%) [25], Inner Mongolia (36.0%) [30], Guangxi (18.10%) [31], but higher than in Guizhou (10.33%) [25], Shanxi (8.92%) [32], Yunnan (12.90%) [33].

The annual prevalence in Chengdu fluctuated between 20.87% and 24.98%. The highest incidence was observed in 2018, followed by a decrease to a relatively stable level. This finding is consistent with a previous study conducted in Shanghai, where the trend of HPV prevalence increased during the period from 2011 to 2018 [26]. However, studies conducted in southern China and Wenzhou showed declined prevalence from 2012 to 2018 [11, 12]. The annual prevalence in Aba has shown an upward trend, increasing from 8.75% to 17.89% (2018–2021), which was consistent with the growth in the screening population.

Understanding the prevalence and distribution of HPV genotypes can facilitate the development and implementation of effective vaccination programs. According to a report, HPV-16, HPV-18, HPV-52, HPV-31, and HPV-58 were the five most prevalent forms globally. In contrast, HPV-16, HPV-18, HPV-52, HPV-51, and HPV-58 were the most prevalent types in Asia [17]. Our study found that the top five HR-HPV genotypes were HPV52, -16, -58, -53, -51 and the top three LR-HPV genotypes were HPV81, -42, -43 in Chengdu, meanwhile HR-HPV52, -53, -16, -18, -58 and LR-HPV CP8304, -43, -6 in Aba. In China, regardless of the genotype that appears first, HPV16, -52, and -58 consistently ranked as the top three [24, 26, 34]. The distribution pattern in Chengdu was similar to this trend. However, in Aba, HPV58 falls to fifth place. According to reports, HPV52 and -58 were the predominant genotypes in Asia, particularly in China, and their infection may be associated with cervical carcinogenesis [35, 36]. Around the world, HPV16 and -18 were the most prevalent genotypes, accounting for up to 70% of cervical malignancies [37]. In our study, HPV16 ranked second, while HPV18 was only ninth in Chengdu. In Aba, HPV16 and -18 were ranked third and fourth. It was noteworthy that among HR-HPV infections, HPV53 was one of the top four in Chengdu and the top two in Aba, which was consistent with some related studies [11, 25]. Moreover, it has been demonstrated that HPV53, a historically non-vaccinated genotype, was associated with the theoretical potency of viral carcinogenicity [38, 39]; therefore, prophylactic vaccines covering HPV53 may provide more comprehensive protection for women in these regions. Additionally, vaccines against HPV51- the fifth prevalent type in Chengdu- should also be considered. Furthermore, our investigation revealed that HPV81, which exhibits similar prevalence as reported in other parts of China, was the most common low-risk HPV type in Chengdu [40, 41]. HPV CP8304 was revealed to be the predominant low-risk genotype in Aba, aligning with Le et al.'s findings [21]. As a result, focused research and development of vaccines will enhance cost-effective vaccination for the Chinese population to a greater extent.

In our study, the most prevalent forms of infection were single-type and high-risk HPV infection in both Chengdu and Aba. Among multiple subtypes, co-infections with two HPV types were the most prevalent, which was consistent with some regional results [25, 42, 43]. Up until now, there is no consensus on whether multiple infection increases cervical cancer risk compared to single-type infection. Multiple infection has been linked to a higher risk of cervical cancer development and occurrence than single infection, as indicated by researches [44, 45]. However, other studies have found that in contrast to multiple infection, single HPV infection was associated with a higher chance of developing cervical cancer [46, 47].

The prevalence of HPV infection in Chengdu was highest among the youngest age group and second highest among the oldest. This bimodal tendency was supported by other researches [11, 17]. Notably, HR-HPV infection demonstrated a similar pattern. Previous studies have shown that the "two-peak" pattern exhibited in HR-HPV infection of Chinese women

[48]. Meanwhile, LR-HPV and mixed infection also showed similar distributions. As for the top six HPV genotypes in our study, the highest incidence was observed among individuals aged ≤25 years old, while the second-highest was in those aged ≥46 years old. Additionally, we presented the prevalence of single, dual, and multiple HPV infections stratified by age groups. All displayed a bimodal distribution with peaks at ≤25 and ≥46. This outcome may be attributed to active sexual activity among younger individuals and persistence or reactivation of latent infection among menopausal individuals. These findings suggest a greater need for focused prevention and management of HPV infection in both young and older women.

The study's strengths lie in its substantial sample size, utilization of unmodified HPV genotyping techniques, and a six-year observation period that tracked annual epidemic trends in HPV infection in Chengdu. This was a dearth of data from remote high-altitude localities in the existing research and the epidemiological characteristics of HPV in Aba located in the Tibet Plateau were reported for the first time in our study. This study proved to be a valuable endeavor in assessing HPV prevalence and genotype distribution across diverse regions, thereby facilitating the implementation of vaccination programs. However, our study has several limitations. Firstly, due to the limited sample size, the results obtained in Aba may not comprehensively reflect the epidemiological characteristics of HPV infection. Thus, a follow-up study with an adequate sample size and monitoring period is imperative. Despite the continuous increase in HPV screening participation over the years, the overall number of participants in Aba remains low. To extend screening coverage, public health initiatives aimed at enhancing awareness of HPV screening and prevention should be reinforced and expanded. Secondly, due to the absence of cervical cytology data, it was not feasible to accurately examine the correlation between HPV genotypes and precancerous lesions or cervical cancer. Lastly, owing to the lack of specific patient information, such as education level, history of HPV infection, and HPV vaccination, analysis on how these factors influenced the prevalence of HPV infection could not be conducted.

To summarize, our study has estimated the prevalence of HPV infection rates, annual trends, age-specific prevalence, and type distribution in Chengdu and Aba of Sichuan Province. Based on the age profile of HPV prevalence in the two districts, it is recommended that the age range for free screening be expanded. Our data also showed a high prevalence of some non-vaccine genes, making the development of region-specific vaccines imminent. HPV vaccination programs can protect from HPV infection and then reduce the future burden of invasive cervical cancer, but the current cervical cancer vaccine is in short supply and expensive to administer, so it is hoped that the cost of the vaccine will be covered by insurance or health insurance in the future. These results are directly applicable to local governments for promoting HPV-targeted vaccination in the study regions. It also offers recommendations for vaccination selection and cervical cancer prevention in western China.

## Supporting information

**S1 File.**
(DOCX)

## Acknowledgments

The authors thank everyone who helped in this study

## Author Contributions

**Conceptualization:** Qianqian Wang, Min Xu.

**Data curation:** Qianqian Wang, Min Xu, Yahui Li.

**Investigation:** Yahui Li, Jichun Ma, Weijun He.

**Methodology:** Hua Zhou.

**Project administration:** Hua Zhou.

**Resources:** Jichun Ma, Weijun He.

**Software:** Xuan Zhu.

**Supervision:** Xuan Zhu.

**Writing – original draft:** Qianqian Wang.

**Writing – review & editing:** Min Xu.

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
