## [Decision Letter · Decision Letter 0]

8 Jan 2024

PONE-D-23-41668Prevalence characteristics of cervical human papillomavirus infection in Chengdu and Aba District, Sichuan Province, ChinaPLOS ONE

Dear Dr. WANG,

Thank you for submitting your manuscript to PLOS ONE. After careful consideration, we feel that it has merit but does not fully meet PLOS ONE’s publication criteria as it currently stands. Therefore, we invite you to submit a revised version of the manuscript that addresses the points raised during the review process.

We look forward to receiving your revised manuscript.

Kind regards,

Kazunori Nagasaka

Academic Editor

PLOS ONE

Journal Requirements:

Additional Editor Comments:

Dear Authors,

Thank you very much for your submission to Plos One.

Our decision is a major revision. As one reviewer mentioned in the report, the manuscript is useful for understanding the current situation of HPV infection in China.

However, thinking a novelty, the authors should put more emphasis on their novel findings based on the database and claim some suggestions on how to reduce the burden of cervical cancer in China.

We look forward to receiving the revised manuscript.

Sincerely,

Plos one

Reviewers' comments:

Reviewer's Responses to Questions

**Comments to the Author**

1. Is the manuscript technically sound, and do the data support the conclusions?

Reviewer #1: Yes

Reviewer #2: Yes

2. Has the statistical analysis been performed appropriately and rigorously? 

Reviewer #1: Yes

Reviewer #2: Yes

3. Have the authors made all data underlying the findings in their manuscript fully available?

Reviewer #1: Yes

Reviewer #2: Yes

4. Is the manuscript presented in an intelligible fashion and written in standard English?

Reviewer #1: Yes

Reviewer #2: No

5. Review Comments to the Author

Reviewer #1: This is a well conducted original work on the prevalence and type specific distribution of HIV infections in the two different population of Sichuan Province of China. This gives an insight into the inclusion and development of Vaccine against these current non-vaccine HPV types also.

Suggested to discuss and compare with the distribution of any of these study prevalent types with the regions other than China and other continents also.

Reviewer #2: In the manuscript titled "Prevalence characteristics of cervical human papillomavirus infection in Chengdu and Aba District, Sichuan Province, China", Wang et al. described the prevalence and genotype distribution of HPV in Chengdu and Aba District. The data provided in this study may be helpful for government-level decision-making, but revision is required.

Major comments:

1.The two units mentioned in this study are administratively closely related (one is a subordinate of the other one)? If not, the ethical approval from the second unit is also required.

2.The population included in the two units are possibly different. Participants from the Chengdu hospital tended to be patients (Outpatients) and residents (physical examination) while those from the Aba hospital tended to be from a cancer screening programme (possibly dominated by patients or residents, but not both, and most probably the latter) which could have contributed to the significant difference between the two regions. The authors didn’t mention this point in the discussion section.

3.There are claims not supported by the data provided in the manuscript, such as in line 314-317, the authors concluded the difference in the awareness of preventive measures, but in the manuscript data supporting such claim is not to be found.

4.The genotyping kits used in the study differ in the number of total HPV subtypes covered, which should have contributed to the observed difference in the prevalence, however, the authors didn’t mention its impact to the final results.

5.Is CP8304 genotype 81? Check this out and make corrections.

Minor points

2.The manuscript requires extensive revision to correct for incorrect use of words and inappropriate sentences. To list just a few: line 62-64; line 65-66; line 72-73; line 76-77; line 88; line 108; line 284-286......

6. PLOS authors have the option to publish the peer review history of their article (what does this mean?). If published, this will include your full peer review and any attached files.

Reviewer #1: No

Reviewer #2: **Yes: **Yuqi Huo

---

## [Author Response · Author response to Decision Letter 0]

7 Mar 2024

Special thanks to Editors and Reviewers for their good comments, and we have responded to every question asked by them in a separate document called "Response to Reviewers".

---

## [Decision Letter · Decision Letter 1]

11 Apr 2024

PONE-D-23-41668R1Prevalence characteristics of cervical human papillomavirus infection in Chengdu and Aba District, Sichuan Province, ChinaPLOS ONE

Dear Dr. WANG,

Thank you for submitting your manuscript to PLOS ONE. After careful consideration, we feel that it has merit but does not fully meet PLOS ONE’s publication criteria as it currently stands. Therefore, we invite you to submit a revised version of the manuscript that addresses the points raised during the review process.

We look forward to receiving your revised manuscript.

Kind regards,

Kazunori Nagasaka

Academic Editor

PLOS ONE

Journal Requirements:

**Additional Editor Comments:**

Dear Authors,

Please revise the manuscript according to Reviewer's comment.

Sincerely,

Kazunori Nagasaka

Reviewers' comments:

Reviewer's Responses to Questions

**Comments to the Author**

1. If the authors have adequately addressed your comments raised in a previous round of review and you feel that this manuscript is now acceptable for publication, you may indicate that here to bypass the “Comments to the Author” section, enter your conflict of interest statement in the “Confidential to Editor” section, and submit your "Accept" recommendation.

Reviewer #1: All comments have been addressed

Reviewer #2: All comments have been addressed

2. Is the manuscript technically sound, and do the data support the conclusions?

Reviewer #1: Yes

Reviewer #2: Yes

3. Has the statistical analysis been performed appropriately and rigorously? 

Reviewer #1: Yes

Reviewer #2: Yes

4. Have the authors made all data underlying the findings in their manuscript fully available?

Reviewer #1: Yes

Reviewer #2: Yes

5. Is the manuscript presented in an intelligible fashion and written in standard English?

Reviewer #1: Yes

Reviewer #2: Yes

6. Review Comments to the Author

Reviewer #1: (No Response)

Reviewer #2: The authors have answered most of all my questions. Some minor problems remain.

Minor comments

1.In the Abstract, the final sentence can be changed to '....HPV screening provides valuable guidance......'。

2.Suggest change 'Furthermore, our investigation revealed that HPV81........other parts of China......' to 'Furthermore, our investigation revealed that HPV81, which exhibits similar prevalence as reported in other parts of China, was the most common low-risk HPV type in Chengdu'.

7. PLOS authors have the option to publish the peer review history of their article (what does this mean?). If published, this will include your full peer review and any attached files.

Reviewer #1: No

Reviewer #2: No

---

## [Author Response · Author response to Decision Letter 1]

9 May 2024

We appreciate for Editors'/Reviewers' warm work earnestly and hope that the correction will meet with approval. All responses are in the file "Response to Reviewers". Once again, thank you very much for your comments and suggestions.

---

## [Decision Letter · Decision Letter 2]

20 May 2024

Prevalence characteristics of cervical human papillomavirus infection in Chengdu and Aba District, Sichuan Province, China

PONE-D-23-41668R2

Dear Dr. WANG,

We’re pleased to inform you that your manuscript has been judged scientifically suitable for publication and will be formally accepted for publication once it meets all outstanding technical requirements.

Kind regards,

Kazunori Nagasaka

Academic Editor

PLOS ONE

Additional Editor Comments (optional):

Dear Authors,

Congratulations！

I am pleased to inform you that your manuscript is now acceptable for publication in Plos One.

We look forward to your future manuscript.

Thank you for your submission.

Sincerely,

Kazunori Nagasaka

Reviewers' comments:

Reviewer's Responses to Questions

**Comments to the Author**

1. If the authors have adequately addressed your comments raised in a previous round of review and you feel that this manuscript is now acceptable for publication, you may indicate that here to bypass the “Comments to the Author” section, enter your conflict of interest statement in the “Confidential to Editor” section, and submit your "Accept" recommendation.

Reviewer #1: All comments have been addressed

Reviewer #2: All comments have been addressed

2. Is the manuscript technically sound, and do the data support the conclusions?

Reviewer #1: Yes

Reviewer #2: Yes

3. Has the statistical analysis been performed appropriately and rigorously? 

Reviewer #1: Yes

Reviewer #2: Yes

4. Have the authors made all data underlying the findings in their manuscript fully available?

Reviewer #1: Yes

Reviewer #2: Yes

5. Is the manuscript presented in an intelligible fashion and written in standard English?

Reviewer #1: Yes

Reviewer #2: Yes

6. Review Comments to the Author

Reviewer #1: (No Response)

Reviewer #2: The authors have answered all my questions and i think it is appropriate to accept in its current form.

7. PLOS authors have the option to publish the peer review history of their article (what does this mean?). If published, this will include your full peer review and any attached files.

Reviewer #1: No

Reviewer #2: **Yes: **Yuqi Huo

---

## [Editor Report · Acceptance letter]

4 Jun 2024

PONE-D-23-41668R2 

PLOS ONE

Dear Dr. Wang, 

I'm pleased to inform you that your manuscript has been deemed suitable for publication in PLOS ONE. Congratulations! Your manuscript is now being handed over to our production team.

Kind regards, 

on behalf of

Professor Kazunori Nagasaka 

Academic Editor

PLOS ONE